# How do changes in warm-phase microphysics affect deep convective clouds?

Qian Chen[1,2], Ilan Koren[1*], Orit Altaratz[1], Reuven H. Heiblum[1], Guy Dagan[1] and Lital Pinto[1]

[1]Department of Earth and Planetary Sciences, Weizmann Institute of Science, Rehovot, Israel
[2] School of Atmospheric Physics, Nanjing University of Information Science & Technology, Nanjing, China

*Correspondence to: Ilan Koren (ilan.koren@weizmann.ac.il)

## Abstract

Understanding aerosol effects on deep convective clouds and the derived effects on the radiation budget and rain patterns can largely contribute to estimations of climate uncertainties. The challenge is difficult in part because key microphysical processes in the mixed and cold phases are still not well understood. For deep convective clouds with a warm base, understanding aerosol effects on the warm processes is extremely important as they set the initial and boundary conditions for the cold processes. Therefore, in this study the focus is on the warm phase, which can be better resolved. The main question is: "How do aerosol-derived changes in the warm phase affect the properties of deep convective cloud systems?" To explore this question, we used the weather research and forecasting (WRF) model with spectral bin microphysics to simulate a deep convective cloud system over Marshall Islands during the Kwajalein Experiment (KWAJEX). The model results were validated against observations, showing similarities in the vertical profile of radar reflectivity and the surface rain rate. Simulations with larger aerosol loading resulted in a larger total cloud mass, a larger cloud fraction in the upper levels, and a larger frequency of strong updrafts and rain rates. Enlarged mass both below and above the zero temperature level (ZTL) contributed to the increase in clouds' total mass (water and ice) in the polluted runs. Increased condensation efficiency of cloud droplets governed the gain in mass below the ZTL, while both enhanced condensational and depositional growth led to increased mass above it. The enhanced mass loading above the ZTL acted to reduce the clouds' buoyancy while the thermal buoyancy (driven by the enhanced latent heat release) increased in the polluted runs. The overall effect showed an increased

upward transport (across the ZTL) of liquid water, driven by both larger updrafts and larger droplet mobility.

These aerosol effects were reflected in the larger ratio between the masses located above and below the ZTL in the polluted runs. When comparing the net mass flux crossing the ZTL in the clean and polluted runs, the difference was small. However, when comparing the upward and downward fluxes separately, the increase in aerosol concentration was seen to dramatically increase the fluxes in both directions, indicating the aerosol-amplification effect of the convection and affecting cloud-system properties such

as cloud fraction and rain rate.

## 1. Introduction

The overall effect of aerosol on clouds is one of the most challenging questions in climate research (IPCC, 2013). Within this domain, aerosol interactions with convective clouds and the derived effects on rain patterns are especially difficult to determine due to the tight coupling of dynamic and

40 microphysical processes (Altaratz et al., 2014; Fan et al., 2016; Tao et al., 2012). The environmental thermodynamic conditions determine the overall potential for cloud and precipitation formation, whereas aerosol properties dictate how efficiently the cloud will develop within this given environmental potential (Dagan et al., 2015a,b). Aerosols act as cloud condensation nuclei (CCN) and ice nuclei, thus affecting the initial size distributions of cloud droplets and ice crystals, respectively.

Higher aerosol loading means an increased amount of CCN and therefore activation of more, albeit smaller droplets with a narrower size distribution (Squires and Twomey, 1961; Twomey, 1977; Warner and Twomey, 1967). More activated droplets at the initial stage of a cloud's lifetime will provide a larger collective surface area for condensation and therefore, more efficient consumption of the available supersaturation (Dagan et al., 2015a; Koren et al., 2014; Pinsky et al., 2013; Seiki et al.,

2014). Moreover, efficient condensation in polluted cases is prolonged because the onset of the collision–coalescence process is delayed (Albrecht, 1989; Saleeby et al., 2010; Squires, 1958). This is also reflected in delayed rain onset (Berg et al., 2008; Dagan et al., 2015b; Jin and Shepherd, 2008; Suzuki et al., 2008; Xue et al., 2008).

Deep convective cloud invigoration by aerosols has been shown in observational (Andreae et al., 2004; Koren et al., 2005; Storer et al., 2014) and modeling (Khain et al., 2005; Lee et al., 2008) studies. Special attention has been given to the mixed and cold processes, for which smaller supercooled droplets are likely to freeze at lower temperatures (Rosenfeld and Woodley, 2000). Therefore, the freezing latent heat will be released higher in the atmosphere, further enhancing convection (Han et al., 2012; Koren et al., 2008; Lynn et al., 2007; Rosenfeld et al., 2008; Tao et al., 2007). The intensified convection in the polluted environment increases cloud top height, and enlarges the cloud cover in the upper troposphere due to a larger number of small-size ice crystals at those altitudes that settle slowly (Fan et al., 2013; Koren et al., 2010; Lee and Feingold, 2013; Morrison and Grabowski, 2011; Storer et al., 2014). The larger amount of supercooled droplets has been suggested to be one of the critical elements in the cloud-invigoration chain of events, and it depends on the upward transport of liquid mass across the zero temperature level (ZTL).

Koren et al. (2015) recently showed that an important component of the aerosol effect on clouds can be captured by the effective terminal velocity ($\eta$) properties. $\eta$ is calculated as the mass-weighted-mean hydrometeor terminal velocity within a given volume in the cloud. As such, it describes the terminal velocity of the given volume's center of gravity (COG). In other words, $\eta$ describes the COG velocity (always down) relative to the surrounding air velocity. Smaller droplets with narrow variance will have significantly smaller $|\eta|$ and therefore, in a given updraft, will move higher in the atmosphere than larger droplets (larger $|\eta|$). The sum of the surrounding air velocity ($w$) and $\eta$ describes the hydrometeor's COG velocity with respect to the surface. We define this velocity as $V_{COG} = w + \eta$, and it captures both the aerosol effect on the condensation efficiency (as it controls the latent-heat release that fuels the cloud's updraft, $w$) and the afore-described effect on $\eta$. In the early stages of cloud evolution, when condensation is the dominant process (the collection process is not yet significant), higher condensation efficiency and smaller $|\eta|$ act together to push the hydrometeors higher in the polluted clouds. Such effects have only been shown and discussed for warm clouds (Dagan et al., 2015a; Dagan et al., 2016; Heiblum et al., 2016; Koren et al., 2015; Koren et al., 2014).

A big part of the challenge in understanding aerosol effects on deep convective clouds is attributed to the large uncertainties related to ice-nucleation processes (DeMott et al., 2015). These

processes have been shown to be extremely sensitive to the aerosol surface properties in a manner that is not yet understood (Vali, 2014), and many freezing schemes are based on empirical relationships that do not create one comprehensive theory.

Here we focus on warm-phase processes within a deep convective system. Warm-phase processes (which are less complex) dictate much of the boundary and initial conditions with regard to hydrometeors, moisture and heat fluxes to the mixed and cold phases. More specifically, the objectives of this study are to explore the aerosol effects on the condensate mass fluxes crossing the ZTL to understand the role of warm microphysical processes in deep convective clouds. We also analyze
processes in the mixed and cold phases, as well as downward fluxes crossing the ZTL from the mixed to warm phase, with the caveat of less certainty in the distribution of the specific ice particles.

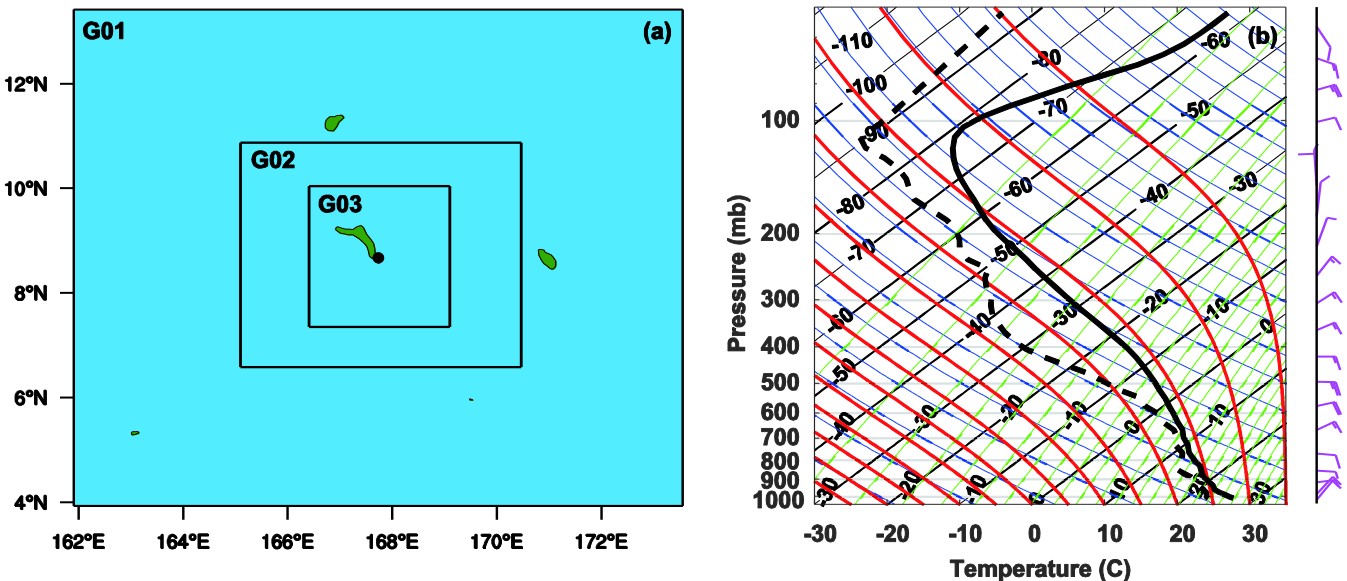

Fig. 1 (a) Three nested domains over the Marshall Islands with horizontal grid spacing of 12 km (G01), 2.4 km (G02), and 0.48 km (G03). (b) Mean vertical profiles of temperature (⁰C) and dew-point temperature (⁰C) at the initial time of the simulations in G03
(1200 UTC, 19 Aug 1999). The black dot in (a) (center of G03) denotes the location of the Kwajalein S-band radar.

## 2. The model and simulation setups

The simulations were conducted using the weather research and forecasting (WRF) model, version 3.6.1, including a fast version of spectral bin microphysics (Fast-SBM) (Khain et al., 2009; Skamarock et al., 2008). Three size distributions of hydrometeors were used to describe water drops, ice crystals/snow and graupel, and each one was defined by 33 mass doubling bins, i.e., the mass of the hydrometeors in bin ($k$) was double the particle mass in bin ($k$-1). CCN were described by a separate size distribution containing 33 bins, with minimum and maximum sizes of 1.23 nm and 2 μm, respectively. More details about the warm and cold processes considered in Fast-SBM can be found in Khain et al. (2004), Khain et al. (2009) and Lynn et al. (2007).

A deep convective cloud system was simulated over the Marshall Islands during the Kwajalein Experiment (KWAJEX, 23 Jul–25 Sep 1999). The simulations were conducted with three nested grids (i.e., G01, G02 and G03). The domain configuration is shown in Figure 1a; the grid sizes were ~1320 km × 1080 km, 600 km × 480 km and 300 km × 300 km with horizontal grid spacing of 12,000 m, 2400 m, and 480 m and time steps of 40 s, 8 s, and 2 s for G01, G02, and G03, respectively. There were 60 vertical levels for each grid with stretched spacing between 70 m near the ground to 400 m above 2000 m height. The G01 run was driven by the Climate Forecast System Reanalysis (CFSR) data (every 6 h, https://climatedataguide.ucar.edu/climate-data/climate-forecast-system-reanalysis-cfsr) and it provided boundary values for the G02 run using the two-way nested run method. The innermost grid (G03) was driven by the G02 run data every 10 min using one-way nested method (nest-down) to maintain similar initial and boundary conditions in different aerosol-scenario simulations. The G01 and G02 runs were initiated on 19 Aug 1999, 0000 UTC (1200 LT), with spin-up time of 12 h and total run time of 24 h. The G03 run was initiated 12 h later at 1200 UTC (0000 LT) on 19 Aug and ended at 0000 UTC (1200 LT) on 20 Aug. Taking a spin-up time of 4 h, the simulation results for G03 were analyzed between the 4 and 12 h of the simulation. The same configuration of physical schemes was used for all three nested grids including the Fast-SBM microphysical scheme, the RRTMG longwave and shortwave radiation

schemes (Iacono et al., 2008), the Yonsei University (YSU) planetary boundary layer scheme (Hong et al., 2006), and the Noah land surface scheme (Chen and Dudhia, 2001).

The initial CCN distribution in SBM was calculated using the empirical equation $N_{CCN} = N_0 S^k$, where $N_{CCN}$ is the numerical concentration of activated aerosol particles at supersaturation ($S$) with respect to water (%). $N_0$ and $k$ are constants for determining the aerosol concentrations in different aerosol scenarios. The calculation method was as detailed by Khain et al. (2000). Observations showed that the KWAJEX clouds developed in a pristine environment with an average drop number of less than 100 cm$^{-3}$ (Rangno and Hobbs, 2005). Therefore, in the G01 and G02 simulations that produced the meteorological conditions for the G03 run, we used $N_0 = 100$ cm$^{-3}$ and $k = 0.5$. The G03 simulations were carried out for clean ($N_0 = 100$ cm$^{-3}$ and $k = 0.5$), semi-polluted ($N_0 = 500$ cm$^{-3}$ and $k = 0.5$), and polluted ($N_0 = 2000$ cm$^{-3}$ and $k = 0.5$) conditions. The aerosols were initially assumed to be homogeneously distributed in the lowest 2 km layer of the domain, with an exponential decrease with height above this layer. The initial domain's mean profiles of temperature and dew-point temperature for the G03 run for all three cases are shown in Figure 1b, revealing a warm and humid environment that supports the formation and development of deep convection with a maximum relative humidity of 88% at around 500 m above sea surface level. As shown in Figure 1b, the wind field was dominated by easterly winds; therefore, we excluded a belt of ~50 km near the eastern boundary of G03 from the analyzed region to minimize the impact of cloud formation at the boundary. A narrow belt near the western side (~5 km) of G03 was also excluded from the analyzed data, as well as ~30 km belts near the southern and northern sides of the domain. Results for all three runs were collected every 2 min of simulation time.

In the analysis, we selected 0°C as our reference level for separating the warm and mixed-phase parts in the clouds. Although cloud droplets freeze at colder than 0°C temperatures, above the ZTL there is potential for freezing.

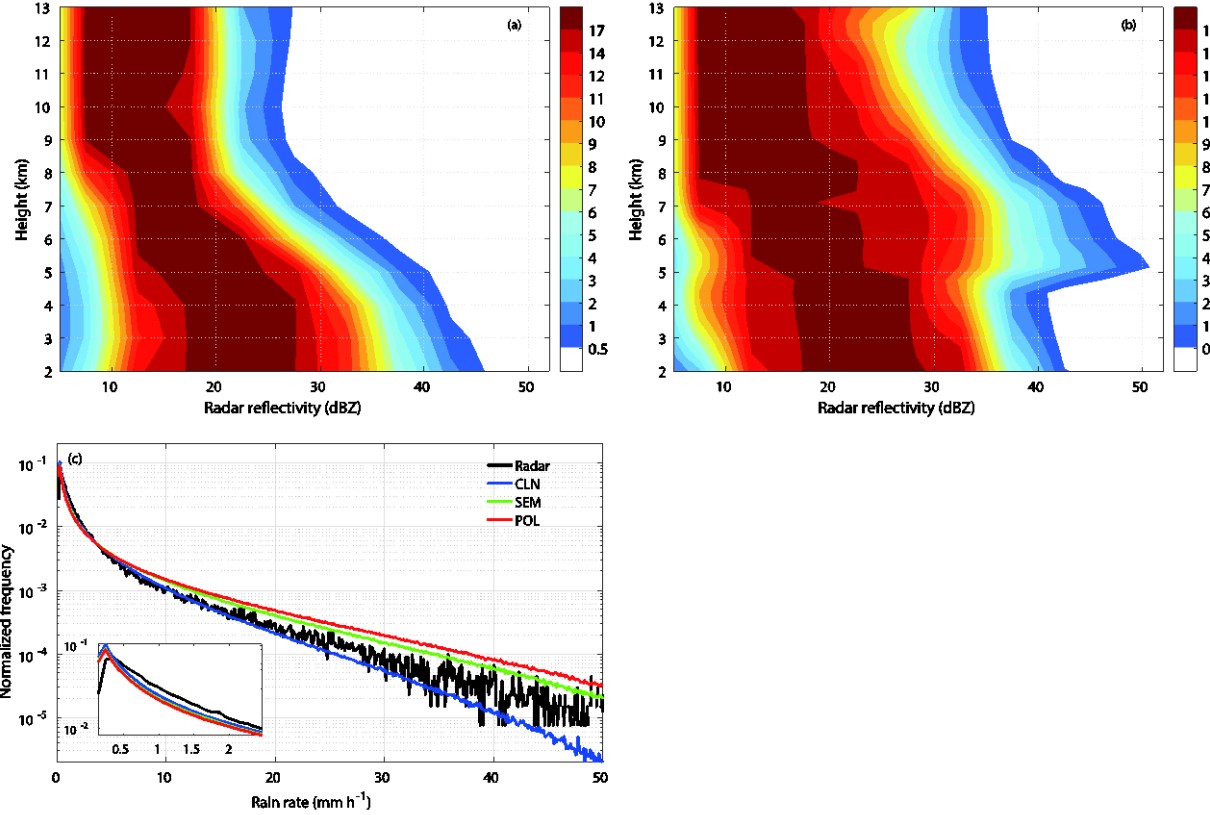

**Fig. 2 Contoured frequency by altitude diagram (CFAD) of reflectivity from (a) Kwajalein S-band radar, (b) clean simulation from 19 Aug 1999, 1600 UTC to 20 Aug 1999, 0000 UTC. (c) Normalized occurrence frequencies of radar-estimated rain rate and modeled results during a similar period (corresponding to 4–12 h of simulation). Only areas with significant rainfall (≥0.15 mm h$^{-1}$) were considered in (c). The figure in the lower left corner shows a zoom-in view of normalized frequency of the small rain rates. 'CLN', 'SEM', and 'POL' stand for clean, semi-polluted, and polluted runs respectively.**

## 3. Results

### 3.1 Comparison with observations

For validation purposes, ground-based Kwajalein S-band radar measurements were used (located as denoted by the black dot in Figure 1a), with 10.71 cm wavelength and a coverage radius of 150 km. A detailed description of the radar measurements in the KWAJEX can be found in Yuter et al. (2005). Figure 2a,b shows a comparison of the contoured frequency by altitude diagrams (CFADs) of

the reflectivity measured by the radar vs. that simulated by the clean run (for 10 cm wavelength).
CFADs are probability-distribution functions, per height level, presented in percentage (Yuter and Houze, 1995). The comparison shows that the clean run captured the vertical structure and magnitude of the observed CFAD reasonably well. The highest probability in the clean run CFAD is located around values of 18–28 dBZ, from the surface up to 4.8 km; above this (5–8 km) it is 12–22 dBZ, and at the upper levels (9 to 13 km) it is between 7 and 17 dBZ, in agreement with the observed radar reflectivity. There is an overestimation of the modeled reflectivity above the ZTL (4.8 km) compared to the observed one. It can be explained by an overestimation of large ice hydrometeors (mostly graupel, but snow particles as well) above the ZTL. This is due to feedbacks caused by the simple melting scheme used by the model (see section 3.3).

Figure 2c shows the normalized frequency of rain rates for 1600–2400 UTC from observations (radar-based estimation as described by Houze Jr et al., 2004) and from the small grids in the three simulations (clean, semi-polluted and polluted). The distributions show a peak in light rain rates (< 0.5 mm h$^{-1}$, zoom-in view in the lower left corner of Figure 2c) and a long tail at the heavy rainfall end. The radar-observed rain-rate distribution is located between the clean and semi-polluted simulation results. Given the uncertainties in rain estimations from radar observations and the model's limitations, this shows that the model captured the general rain-rate distribution.

## 3.2 Aerosol effects on clouds' macrophysical properties

Examination of cloud properties in the three different runs revealed differences in both macrophysical and microphysical properties. Figure 3a,c,e shows the evolution of the vertical profiles of cloud fraction for the three runs (calculated as the ratio between the number of cloudy volume pixels (voxels), i.e., total condensate exceeding 0.01 g kg$^{-1}$, at each vertical level and the total horizontal number of voxels). There is a significantly larger cloud fraction in the middle and upper levels in the semi-polluted and polluted cases compared to the clean run. This trend can be recognized above ~4.8 km and it is very pronounced at the high levels (above 10 km), after the 3$^{rd}$ hour of the simulation. This figure also indicates a higher cloud top height under polluted conditions. Examination of the low-altitude levels (below 4.8 km) also shows a larger cloud fraction in the polluted case, which is probably

the combined effect of mass created in this layer and downward sedimentation of the cloud mass. We note that the cloud fraction below 1 km includes precipitating regions. Figure 3g displays the mean vertical profiles of the cloud fraction from 4 to 12 h of the simulation. It summarizes the trends discussed above.

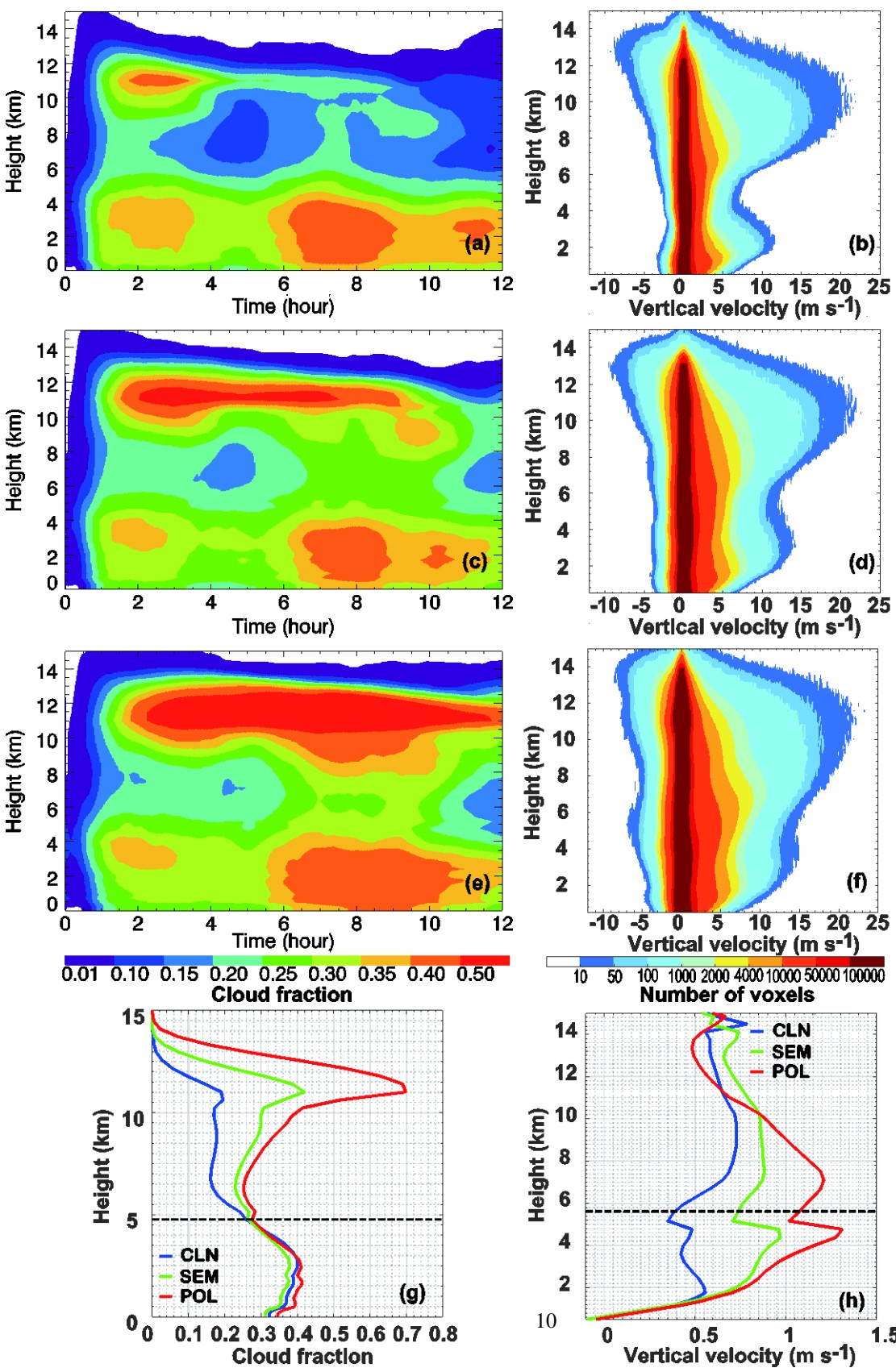

**Fig. 3 (a) Time series of the vertical profiles of a cloud fraction. (b) Frequency of air vertical velocity per altitude from the clean simulation results. (c, d) and (e, f) are similar to (a, b) for the semi-polluted and polluted simulations, respectively. (g) and (h) are vertical profiles of a mean cloud fraction and mass-weighted mean vertical velocity from 4 to 12 h of the simulation in all cases.**

Aside from the larger cloud fraction, Figure 3 also shows stronger updrafts under higher aerosol-loading conditions. Figure 3b,d,f displays the number of voxels (at each altitude) per given interval of vertical velocities, for the clean, semi-polluted, and polluted simulations, respectively. Most of the

200 cloudy voxels are characterized by vertical velocities in the range of -1 to ~1 m s$^{-1}$ in all three runs. The peak updrafts show a bimodal structure with peaks at 2–3 km and at 10–12 km, consistent with the findings of Heymsfield et al. (2010).

The enhanced aerosol loading leads to enlarged areas of both strong updrafts and strong downdrafts from the surface to the upper troposphere. The numbers of voxels with updrafts exceeding 5

205 m s$^{-1}$ increases above the 4.8-km altitude, with an up to 24- and 76-fold increase in the semi-polluted and polluted runs, respectively, compared to the clean run. Figure 3h shows an increasing trend in the mean vertical velocity profiles for more polluted runs, indicating the potential to promote more water vapor and condensate rise to higher altitudes, and hence enhance the growth of hydrometeors at those levels, as shown further on.

Our results agree with previous numerical studies that reported an aerosol invigoration effect of tropical deep convective clouds (Cui et al., 2011; Fan et al., 2013; Khain et al., 2008; Li et al., 2013; Tao and Li, 2016; Tao et al., 2007). However other numerical studies showed no clear evidence for this effect or even an opposite effect (Lee and Feingold, 2010; Morrison and Grabowski, 2011, 2013). The reasons behind those differences were examined in previous studies that showed the lower sensitivity of

cloud and rain processes in bulk schemes to aerosol concentration (Khain et al., 2009, 2015; Lebo and Seinfeld, 2011; Lebo et al., 2012; Heiblum et al., 2016).

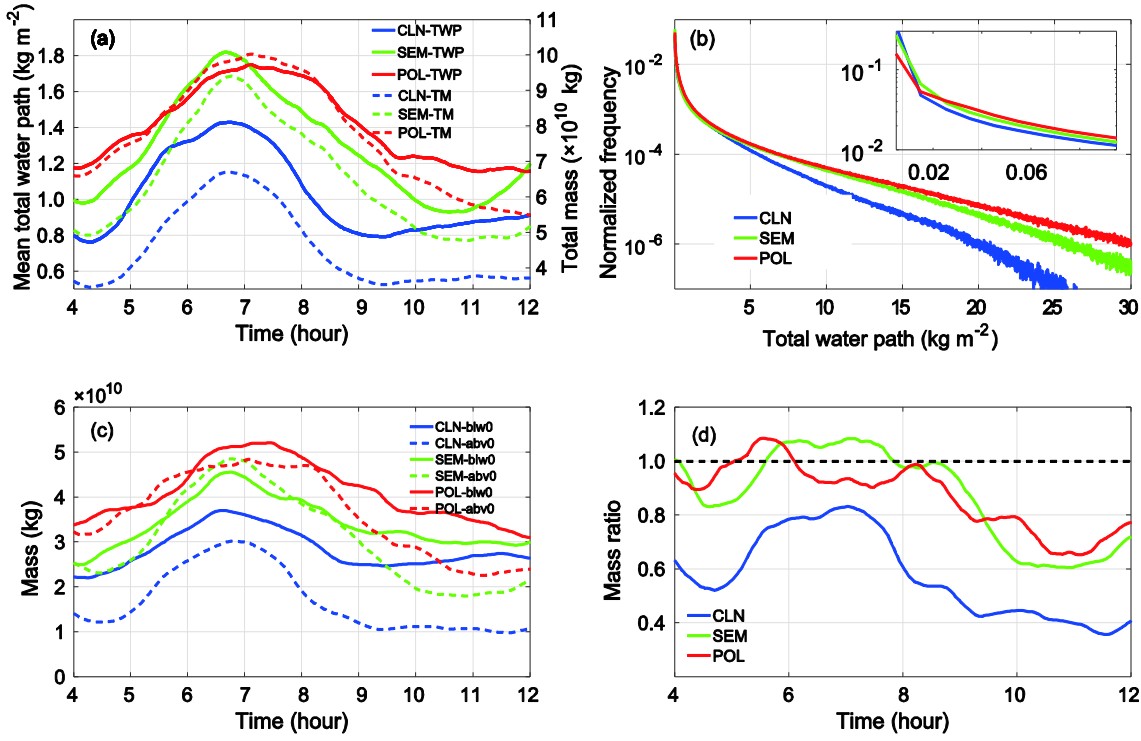

**Fig. 4 (a) Time series of the mean total water path (TWP, kg m⁻²) over the cloudy area (solid curves) and the total cloud mass (kg, dashed curves). (b) Normalized occurrence-frequency distribution of TWP over the domain during 4 to 12 h of the simulation. Time series of (c) total cloud mass above (dashed curves) and below (solid curves) the ZTL and (d) the mass ratios for clean (blue curves), semi-polluted (green curves), and polluted (red curves) simulations.**

Figure 4a shows the time series of the mean total water path (TWP, averaged over cloudy columns) in the clean, semi-polluted, and polluted runs (solid curves). The TWPs in all three runs started increasing at around 4 h into the simulation and peaked between 6 and 7 h of the simulation. The semi-polluted curve shows the highest values (between ~5.5 and 7 h of the simulation) compared with the other two simulations. On the other hand, the total cloud mass (dashed) curves show the highest values for the polluted run and the lowest for the clean run, throughout the simulation. This difference can be explained by the partitioning of the total mass into clouds in the three runs. Even though the total mass was higher in the polluted run it was distributed over a larger area (higher cloud fraction, larger anvils, see Figure 3c,e), and therefore the mean TWP was lower than in the semi-polluted run.

Figure 4b illustrates the normalized occurrence-frequency distribution of the TWP in the three runs. It shows a higher probability for high TWPs as the aerosol amount increases, suggesting that the fraction of deeper clouds (with more integrated mass) was larger in the more polluted runs. The clean run shows a higher frequency of occurrence for the range of small TWPs ($< 0.02$ kg m$^{-2}$, as denoted in the lower left corner of Figure 4b).

To further explore aerosol effects on the vertical distribution of cloud mass, we examined the domain partitioning of total mass in the layers above (in height units, meaning at colder temperatures) and below the ZTL (Figure 4c). It was seen that bigger mass both below and above the ZTL contributes to the largest total mass in the more polluted runs. Exceptionally large was the cloud mass above the ZTL in the semi-polluted run between 6.5 and 7 h of the simulation, which is consistent with the largest mean TWP in this run during that time (Figure 4a). For the clean case, the cloud mass above the ZTL was always smaller than the one below that level. But for the semi-polluted and polluted cases, for short periods, the mass above the ZTL was larger (Figure 4c). As shown in Figure 4d, the ratios between the cloud masses above and below the ZTL were larger in the more polluted cases, suggesting much more efficient upward transport of cloud mass from the warm environment to the subzero temperature region as well as subsequent productive growth of condensate in those levels.

### 3.3 Aerosol effects on clouds' microphysical properties

To gain a better process-level understanding of the aerosol effect on the cloud mass spatial distribution, we evaluated the magnitudes of the phase-transition processes above and below the ZTL. Figure 5 shows the changes in cloud mass (summed for the whole domain between 4 and 12 h of the simulation) driven by the following processes: condensation and evaporation of liquid drops, deposition and sublimation of ice particles, drop freezing and riming by ice particles. Figure 6a-f completes the picture by presenting the total mass per height of the different types of cloud particles.

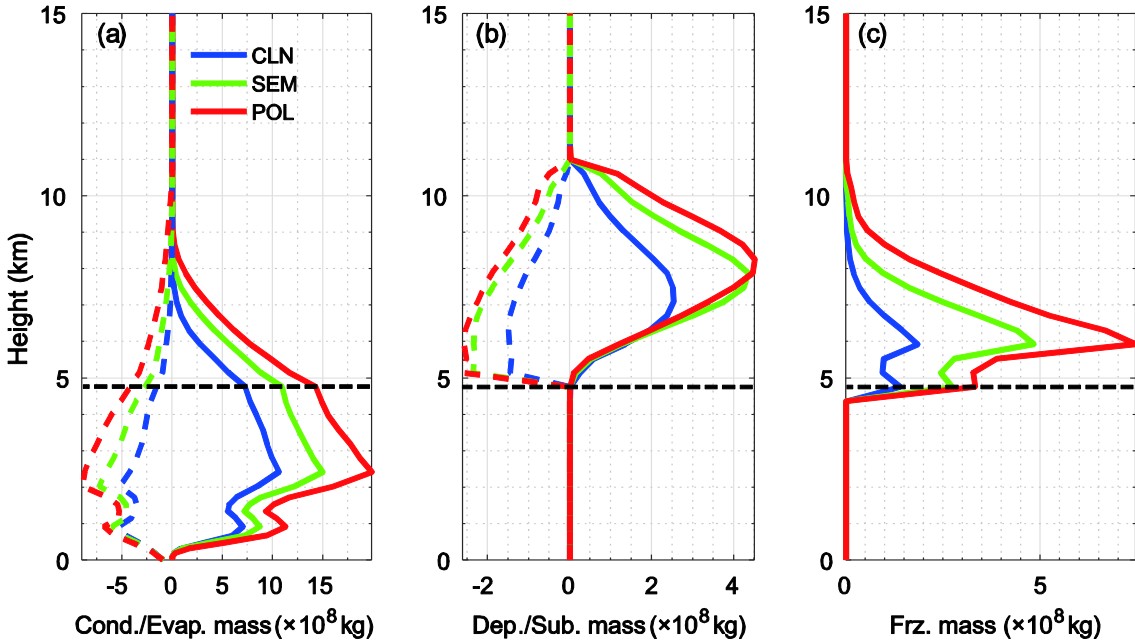

**Fig. 5** Total changes in cloud mass (kg) over 4 to 12 h of the simulation due to (a) condensation/evaporation (solid/dashed curves), (b) deposition/sublimation (solid/dashed curves), and (c) freezing for clean (blue curves), semi-polluted (green curves), and polluted (red curves) experiments. The freezing mass includes both freezing of liquid drops and riming by ice particles. Note that the x-axis scales are different. The dashed black lines denote the zero temperature level (ZTL, ~4.8 km).

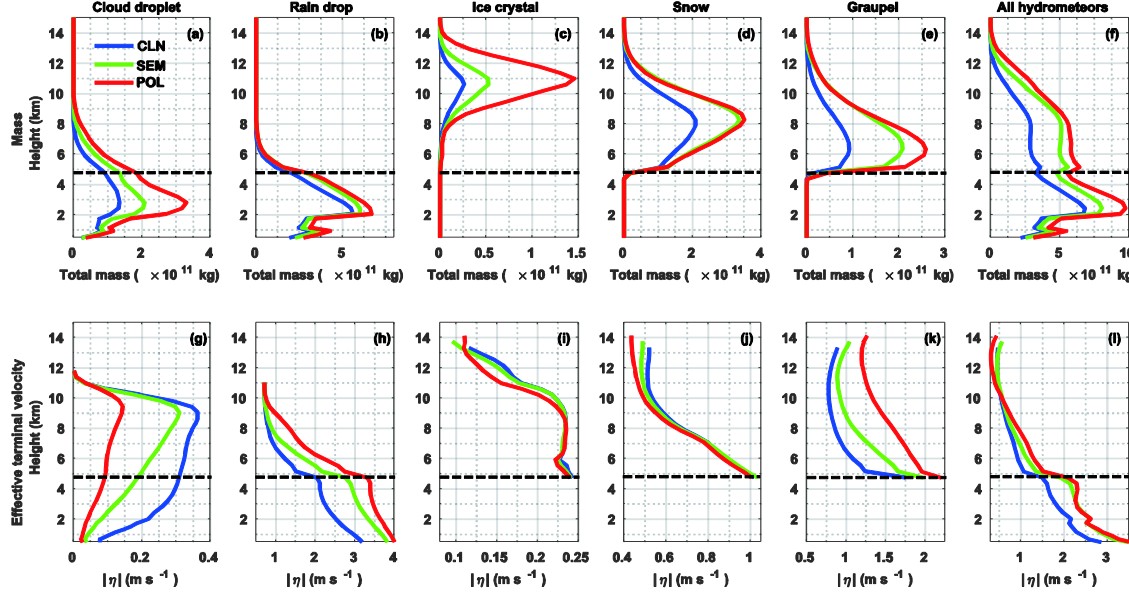

**Fig. 6** Vertical profile of (a–f) total mass over the domain (kg) and (g–l) mean effective terminal velocity ($\eta$, m s$^{-1}$) over cloudy regions during 4-12 h of the simulation for cloud droplet (1$^{st}$ column), raindrop (2$^{nd}$ column), ice crystal (3$^{nd}$ column), snow (4$^{th}$ column), graupel (5$^{th}$ column), and all hydrometeors (6$^{th}$ column). The dashed black lines denote the ZTL (~4.8 km).

The results show an increase in aerosol loading enhanced all of the examined processes, including gain of water and ice mass (condensation and deposition), loss of mass (evaporation and sublimation) and transition from liquid to ice phase (freezing and riming). Since increased aerosol concentration enhanced both types of processes, i.e., both source and sink for the hydrometeor mass, the overall effect was somewhat reduced due to mild cancellation of the gained mass by the enhancement of evaporation and sublimation. Hence, the released and absorbed latent heat were both higher in the more polluted runs, a fact that can explain the enhanced aspect of the dynamics, i.e., stronger updrafts and downdrafts (Figures 3b,d,f). The contribution of the freezing and riming processes to the total latent heat release increased from 4.5% in the clean run to 10.5% in the polluted run, suggesting a larger mass of supercooled water above the ZTL (Figure 6f). The supercooled water reached higher altitudes in the polluted runs (compared to the clean run) as inferred from the higher maximal altitude at which the condensation and freezing processes took place (Figure 5a,c). This is consistent with previous studies showing a higher probability of droplets reaching subzero temperatures under polluted conditions (Carrió and Cotton, 2011; Khain et al., 2001). The freezing/riming of a larger amount of supercooled water and the enhanced depositional growth of ice particles both contributed to the larger ice mass above the ZTL (Figure 6c–e). Note that the simple melting scheme used by the model allowed immediate melting of ice particles while crossing the ZTL. The resulted drops formed by the melting of big graupel (and snow) particles broke up immediately into smaller drops and part of them was carried up again by the updraft and froze by riming. So there may be an overestimation of big grauple particles above the ZTL (as shown in Figure 2b).

The enhanced gain of water and ice mass in the polluted runs yield higher mass loading that acted to reduce the clouds' buoyancy. The upper panel of Figure 7 shows the vertical profiles of the mean buoyancy (total B and components: BT – thermal, BV – vapor and WL – water loading) for the domains' cloudy voxels (between 4 and 12 h of the simulation). Indeed, the water loading played an important role and as expected there was a 'payment', once the polluted clouds got thicker with more

gained liquid and ice mass that was transported higher in the atmosphere, the added water loading acted

to counter-balance the overall buoyancy. Figure 7a shows that the total B profiles switch between smaller values (more negative) for the polluted runs compared to the clean run in low altitudes (from 2.4 to 6.7 km) to larger values at higher levels (above 6.7 km). In all levels, BT was larger in the polluted case (or equal near the freezing level) and the water loading was the smallest (less negative WL buoyancy component) in the clean run, from 1.5 to 8.5 km. The lower panel of Figure 7 shows the vertical profiles of the mean water loading components for the different types of cloud particles. Note, it is different from the profiles of total mass shown in Figure 6 as it presents the mean B values and hence it is influenced by the cloud coverage in each level. It shows significant increase in the water loading (smaller WL buoyancy component) for the polluted runs in all but the snow hydrometeors. Moreover, it reveals that larger rain content, that was likely to originate higher in the clouds can explain about half of the extra water loading in the polluted cases in the lower part of the clouds.

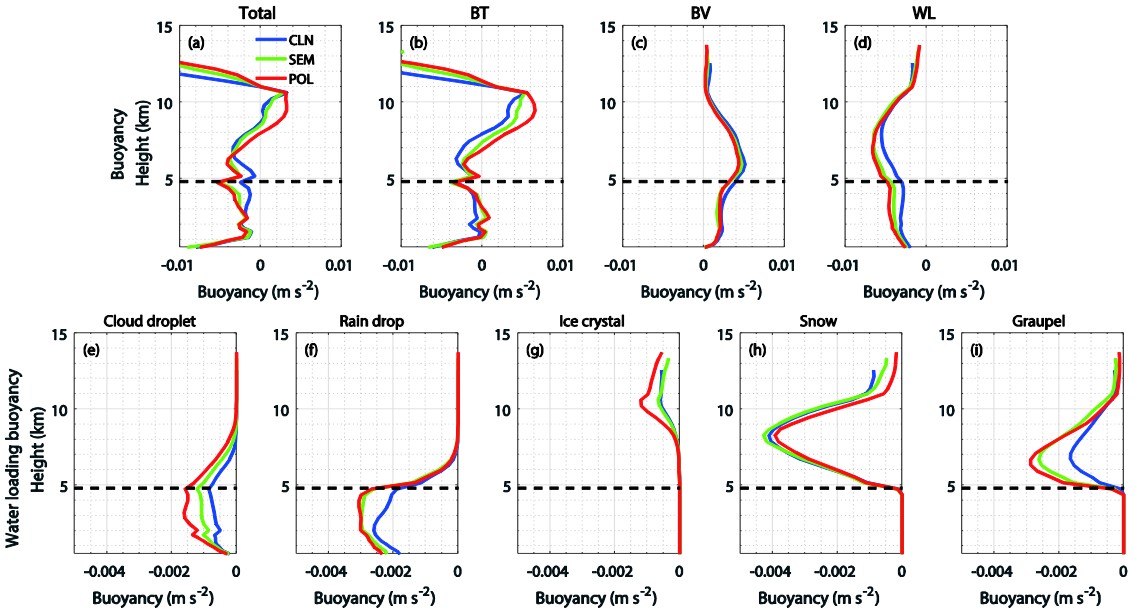

**Fig. 7 Vertical profiles of (a) mean buoyancy averaged over cloudy regions between 4-12 hours of simulation and its components: (b) thermal term (BT), (c) water vapor term (BV), and (d) water loading term (WL); (e-i) vertical profiles of WL for cloud droplet (e), raindrop (f), ice crystal (g), snow (h), graupel (i). The black dashed lines denote the ZTL (~4.8 km).**

It is important to note that among all of the processes, condensation contributed most to the mass gain. It peaked at 2.4 km for all three runs. Moreover, even above the ZTL, condensation still dominated the cloud mass gain compared to depositional growth, for all three runs (Figure 5a vs. 5b). Since the droplet-nucleation process was negligible above the ZTL in the present study (accounting for ~5% of the total nucleated drop mass), most of the cloud drops that grew by condensation above the ZTL originated in the warm part of the clouds. The mass gained by the enhanced condensation in the polluted runs was further boosted by the stronger updrafts that increased the supersaturation. Moreover, having better droplet mobility (see Figure 6g, panel g) further implies that a significant part of the enlarged liquid mass generated below the ZTL in the more polluted cases (as shown in Figure 6a,b) was transported upward by the stronger updrafts (Figure 3h) and continued growing via condensation at altitudes above the ZTL.

### 3.4 Aerosol effects on upward transport of condensate mass

As noted in the previous section, an important part of the aerosols' effect on deep convective clouds has to do with their influence on the transport of mass from the warm part to the upper levels. To evaluate the hydrometeor fluxes, their terminal velocities (which are correlated with their sizes) must be considered. As described above, the effective terminal velocity ($\eta$) of a given volume within a cloud is an measure of the terminal velocity of the volume's hydrometeor COG (Koren et al., 2015). As such, it can be linearly added to the mean air updraft velocity weighted by hydrometeor mass, to infer the hydrometeor's $V_{COG}$ which is the COG velocity relative to the surface.

Since upward motion is considered positive, the sign of $\eta$ is always negative. To avoid confusion, we will discuss the magnitude (absolute values) of $\eta$ hereafter. Figure 6g–l shows the vertical profiles of the mean $|\eta|$ for all types of hydrometeors. The polluted case has the smallest $|\eta|$ for the cloud droplets (Figure 6g), allowing for the water droplets to be pushed higher in the atmosphere by the enhanced updrafts. Raindrops, however, are larger for the polluted case (Figure 6h). We note that the model uses a simple melting scheme in which there is immediate melting when crossing the ZTL downward. Therefore, all of the hydrometeors below this level are liquid only. The $|\eta|$ value of all liquid drops below the ZTL was larger in more polluted cases (Figure 6l); that of graupel particles increased

with more aerosols, similar to raindrops (Figure 6k); finally, that of ice crystals and snow decreased with aerosol loading above 10 km (Figure 6i), suggesting that the larger cloud fraction of the polluted
runs, in the upper troposphere, contains smaller ice particles that exhibit slower sedimentation (Figure 3g and 6l).

What roles do the two characteristic velocities play in the overall aerosol effect? The COG perspective allows calculating the vertical movement of the hydrometeors as a superposition of the two characteristic velocities of the system, i.e., the air vertical velocity (w) and $\eta$. As explained earlier, the
345 sum of the two (weighted by mass) velocities ($V_{COG}$) is the COG velocity relative to the surface (Koren et al., 2015). Using the definition of COG velocity, we calculated the condensate mass fluxes (kg s$^{-1}$) as the product of the total condensate mass and $V_{COG}$ (m s$^{-1}$). Figure 8 shows the temporal evolution of the two average characteristic velocities ($V_{COG}$—the air vertical velocity (w) and $\eta$). We separate pixels of upward and downward mass flux motion by the sign of $V_{COG}$.

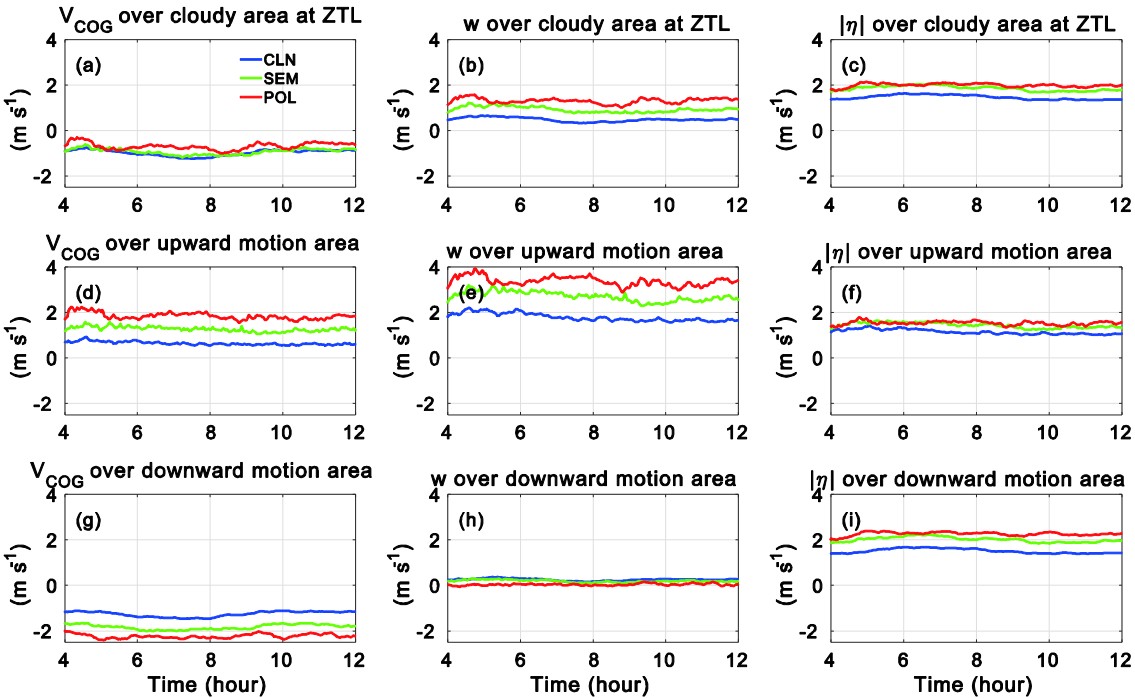

Fig. 8 Time series of (a) mean COG velocity ($V_{COG}$), (b) air vertical velocity ($w$), and (c) effective terminal velocity ($|\eta|$) weighted by mass over a cloudy region at the ZTL; (d–f) and (g–i) are similar to (a–c), but averaged over positive and negative $V_{COG}$ regions, respectively.

We note that despite the fact that the $|\eta|$ value of the cloud droplets is significantly smaller for the polluted clouds (as shown in Figure 6g), allowing them to be pushed higher in the atmosphere by the enhanced updrafts, at the ZTL (~4.8 km), $|\eta|$ values of the polluted hydrometeors are larger than those of the clean ones. This indicates that the contribution of the larger raindrops in the polluted cases controls the overall $|\eta|$ values at this level. For all cloudy area grid boxes (Figure 8a–c), the overall $V_{COG}$ is negative despite the fact that the averaged updrafts are positive, indicating that sedimentation measured by $|\eta|$ values is larger than the updrafts. The positive upward-flux areas (Figure 8d–f) are defined as $V_{COG} > 0$, which implies that the average updrafts have to be larger than $|\eta|$. In contrast, in the negative flux area (Figure 8g–i), the updrafts are fairly weak and $V_{COG}$ is controlled by $|\eta|$.

Finally, we link mass and velocity trends to see the aerosol effect on the total mass flux. Figure 9 shows the temporal evolution of the mass fluxes crossing the ZTL. Similar to the velocity analyses, we separated pixels of upward and downward mass-flux motion by the sign of $V_{COG}$. The domain average total (net) upward and downward fluxes (left column) and the corresponding fraction of the area that represents these fluxes (right column) are shown.

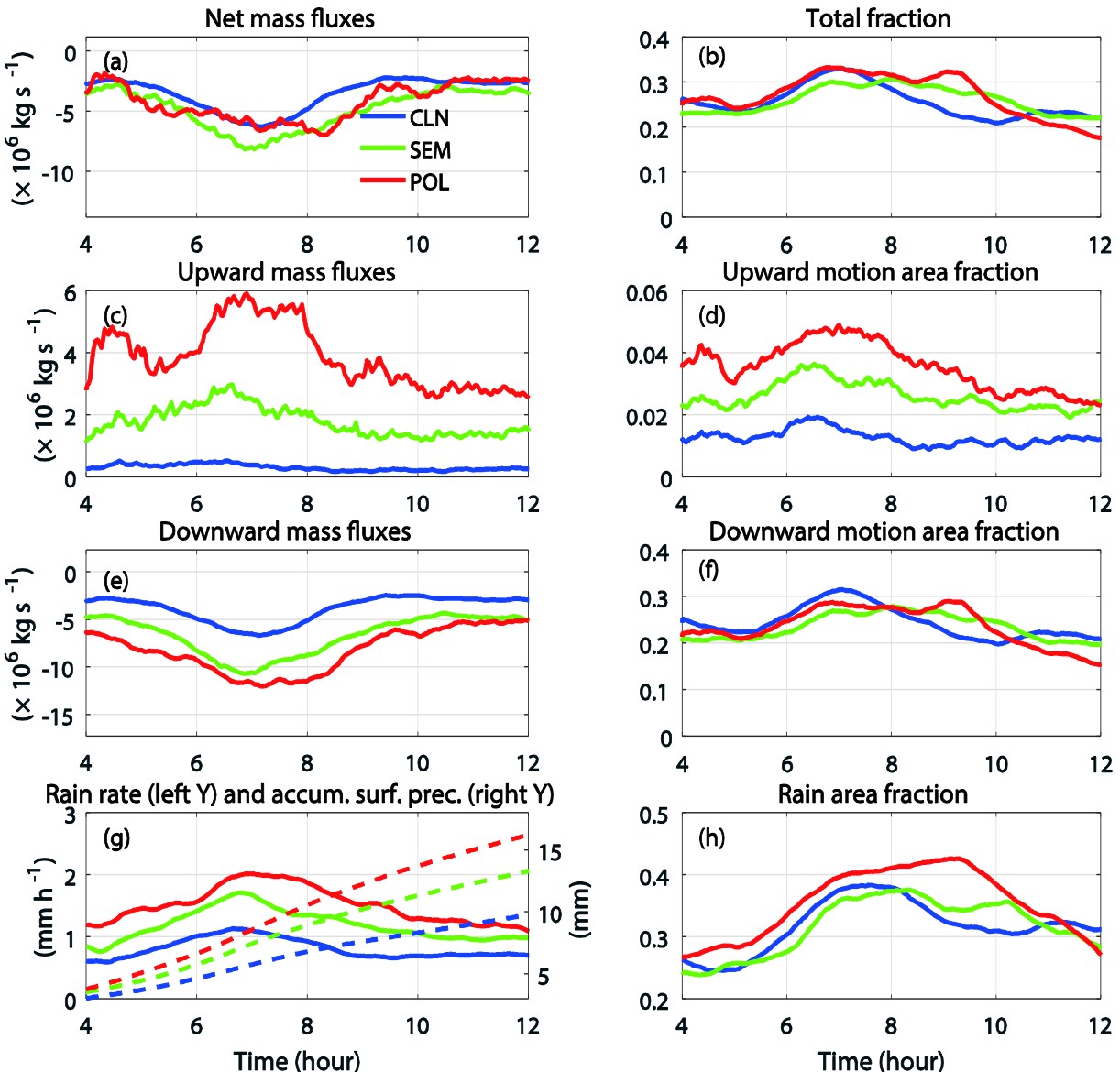

**Fig. 9** Time series of (a) net cloud mass fluxes, and (b) area fraction over entire cloudy area crossing the ZTL for clean (blue curve), semi-polluted (green curve), and polluted (red curve) experiments. (c, d) and (e, f) are similar to (a, b), but for upward ($V_{COG} > 0$) and downward ($V_{COG} < 0$) motion areas, respectively. (g) is time evolution of domain-averaged rain rate (solid lines) and accumulated surface rain (dashed lines). (h) is time evolution of rainy area fraction at the surface.

Similar to the $V_{COG}$ trend, the total (net) mass flux crossing the ZTL (Figure 9a) shows all negative flux values indicating net downward movement of condensate, which is a result of larger mass production above the freezing level and thus sedimentation down. While the temporal evolution of the overall fluxes crossing the ZTL is not dramatically different for the three runs (mean values of ~4.17, 4.77 and $3.46\times10^6$ kg s$^{-1}$ for the polluted, semi-polluted, and clean run, respectively), when the fluxes are observed separately according to their sign, a different view emerges. All flux trends (up and down) were dramatically enhanced in the polluted cases. The upward liquid mass fluxes in the semi-polluted and polluted cases were 4–9 and 8–21 times larger than those in the clean simulation (with mean values of 3.9, 1.8 and $0.32\times10^6$ kg s$^{-1}$ for the polluted, semi-polluted, and clean run). These trends are controlled by the enlarged upward motion area (as shown in Figure 9d), the larger COG velocity (Figure 8d), and the increased liquid water loading. These findings clearly demonstrate the enhanced cloud mass transport to upper levels under more polluted conditions. What is the average ratio between the mass that was transported up from below the ZTL and the mass that was produced locally above the ZTL, in the different runs? This mass ratio is defined as $\mu = \frac{\text{transported mass from below the ZTL}}{\text{mass locally produced above the ZTL}}$ and it is evaluated using the mass fluxes (as shown in Figure 9). If we consider mean values over 8 h of simulation, for which changes in the mass above the ZTL are averaged out, we can use the domain's average mass fluxes across the ZTL to estimate the mass ratio as $\mu \approx \frac{\text{mass flux upward}}{\text{net mass flux}}$. The mass-fraction $\mu$ is much higher in the polluted as compared to the clean run (around 0.94, 0.38 and 0.09 in the polluted, semi-polluted, and clean cases).

We note that for the upward liquid mass transport, the ZTL crossing took place in a relatively small area (an order of magnitude less than the downward motion area, see Figure 9d,f), affecting the mass partitioning below and above the ZTL and thus the formation and growth of ice particles. Moreover, the differences in area for the upward motion between the clean and polluted conditions were the most significant. Figure 9d shows that the upward mass flux area of the polluted case is 1.8–3.9 times larger than for the clean one. This impacted the variance of the mass-flux, which was larger in the more polluted cases (Figure 9c). The increased variance is driven by the enhancement of the fields' dynamics by aerosol, as shown throughout this study. Polluted clouds exhibited larger updrafts with larger

variance (as shown in Figure 8), larger updraft area (Figure 9d) and larger mass fluxes, all of which tend to increase the variance in the upward mass-flux.

Similar to the upward transport, the downward transport of cloud mass from subzero temperature levels to the warm environment (Figure 9e), the mean surface rain rate and accumulated precipitation (Figure 9g) were also larger in the more polluted cases.

## 4. Discussion and Summary

A major fraction of the deep convective clouds around the globe have a warm base. This is more obvious over the tropical belt, where the freezing-level height is located at around 5 km (and the cloud base is at a much lower altitude). This is also true for many of the mid-latitude frontal systems (particularly in the summer). Exceptions to this rule are likely to occur either in a very cold atmosphere, or in the case of orographic clouds for which the lifting condensation level is high.

As such, the warm-phase properties can be considered as the initial and boundary conditions for the mixed and cold phases. Therefore, changes in aerosol loading can affect the mixed- and cold-phase properties not only directly by serving as ice nuclei but also indirectly by affecting the warm-phase properties and the fluxes between the phases. In this work, we focused on the interface between the phases. Although cloud droplets freeze at colder than 0°C temperatures, we selected the ZTL as our reference level between the warm and mixed phases because above it there is a potential for freezing.

We used the WRF model with Fast-SBM to simulate a case study of a deep convective cloud system over the Marshall Islands during the KWAJEX on 19 Aug 1999 (1200–2400 UTC, 19 Aug; 0000–1200 LT, 20 Aug). As a sanity check, we compared the simulated vertical distribution of the radar reflectivity and the normalized occurrence frequencies of radar-estimated rain rates to observations and showed that the model results are similar to the observed values (Figure 2). We analyzed how changes in cloud field properties are related to changes in aerosol concentration, first at the macro level, showing a notable increase in the total cloud mass, a larger cloud fraction in the upper levels, a higher cloud top, and a larger frequency of strong updrafts and heavy rain rates (Figures 3 and 4). Larger mass both below and above the ZTL was shown to contribute to the larger total cloud mass in the polluted runs. Examining processes on a finer scale revealed that increasing aerosol concentration is related to

enhancement of mass (water and ice) source and sink processes (Figure 5). The aerosol effect on the cloud warm-phase processes could be divided into two main branches: one linked to enhancement of diffusion processes (condensation and evaporation) and the other to greater droplet mobility. Per given volume within the cloud, we refer here to droplet mobility as the way in which the COG of the total hydrometeor mass moves with the surrounding air ($w$, updraft) (Koren et al., 2015). The effective terminal velocity ($\eta$) is inversely proportional to the droplet mobility and is the measure for the water mass COG velocity.

More aerosols yield more activated droplets. This implies enhancement of the overall condensation rate that drives more latent heat release (Figure 5a – condensation). At the cloud edges, under subsaturated conditions, the evaporation is enhanced following the same line of reasoning (Figure 5a – evaporation). All of this is in agreement with Lee and Feingold (2013) for deep convective clouds and several studies of warm convection (Dagan et al., 2015a; Dagan et al., 2015b; Koren et al., 2014; Pinsky et al., 2013; Seiki and Nakajima, 2014). The condensation process still played a major role, even above the ZTL, and it made a larger contribution to the gain in mass than did depositional growth. Since droplet activation was negligible above the ZTL, we suggest that the liquid drops participating in the condensational growth come from below the ZTL.

$V_{COG} = w + \eta$ captures both the aerosol effect on the condensation efficiency (as it controls the latent heat release that fuels the cloud's updrafts ($w$)) and the mobility effect as captured by $\eta$ (Koren et al., 2015). We calculated condensate mass fluxes across the ZTL as a product of cloud mass and $V_{COG}$. Again, on the same micro scale, an increase in aerosol concentration was related to flux enhancement in all directions. Although the net mass fluxes changed only slightly, the condensate mass fluxes up and down were dramatically amplified. Larger $V_{COG}$ led to enhanced upward transport of liquid mass from warm to mixed parts under polluted conditions (Figure 9c,d). The overall aerosol effects are reflected by a larger ratio between the cloud mass above and below the ZTL. We show that there is a "payment" for the enhanced mass production and transport in the polluted runs. The larger mass loading act to reduce the buoyancy while the enhanced latent heat release act to increase it. The enhanced mass flux upward and the enhanced mass production above the ZTL yield enhancement of the mass flux down (partly manifested as enhanced rain that explains much of the increase in mass loading below the ZTL).

Our study highlights the importance of aerosol effects on the warm processes in deep convective clouds, using condensate mass flux as a measure of hydrometeor transport in clouds, between the warm and mixed and cold domains. Such effects enhance the thermodynamic and dynamic (vertical winds) processes as well as changes in the overall structure and properties of the field (demonstrated here as cloud fraction per height or changes in rain rate).

**Acknowledgments**

This study was supported by the European Research Council under the European Union's Seventh Framework Program (FP7/2007-2013)/ERC grant agreement 306965. Qian Chen also acknowledges support from the National Key Research and Development Program of China (grant 2017YFA0604000), the National Basic Research Program of China (grant 2014CB441403), the National Science Foundation of
470 China (grant 41405126, 91644224, 41590873), the Public Meteorology Special Foundation of Ministry of Science and Technology of China (grant GYHY201306047), and the Priority Academic Program Development (PAPD) of Jiangsu Higher Education Institution. This research used computing resources on the WEXAC at the Weizmann Institute, Israel, as well as resources on the Milkyway-2 at the National Supercomputer Center in Guangzhou, China.

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
