# Peer review of "How do changes in warm-phase microphysics affect deep convective clouds?"

_Atmospheric Chemistry and Physics, 2016_

## Referee Comment (RC1) · Anonymous Referee #1 · 30 Dec 2016

This manuscript presents an analysis of aerosol effects on tropical deep convection using the WRF model with detailed bin microphysics. It emphasis the zero degree level as explaining the system wide behavior. It adds analysis of the velocity center of gravity (VCOG) to help explain the aerosol invigoration of deep convection, which is a hotly debated topic in the cloud physics/dynamics community.

I have a few comments that I think will make the paper much more persuasive.

1) A counter argument to invigoration is that the added mass loading reduces the buoyancy and offsets the higher condensation/latent heating. Please show analysis (e.g., profile time series) of the components of the buoyancy (thetav' and mass loading) for all cases. Mass loading should be broken down for the different hydrometeors. This is important in explaining why the added mass loading does not seem to make much

difference. Can it be tied to the hydrometeor mobility? This perspective would make the paper much stronger.

2) There is emphasis in the tone of the paper (even the title) on understanding the warm phase and transport of water across the ZTL and how that affects the mixed phase above. But I didn't find this persuasive. E.g., On page 14 it is stated: "most of the cloud drops that grew by condensation above the ZTL originated in the warm part of the clouds. Therefore, we can clearly state that an important part of the enlarged liquid mass generated below the ZTL in the more polluted cases (as shown in Figure 6a,b) was transported upward with the stronger updrafts.." However, a more straightforward interpretation is that the stronger updrafts, which extend above the ZTL, generate more cooling, higher supersaturation and therefore more condensate. The 'transport' argument doesn't seem particularly clear or strong. On this point the figures need some work (see # 9 below). My suggestion is to either strengthen the argument of why a stronger mass flux at ZTL necessarily carries up through the depth of the cloud, or alternatively remove the emphasis.

3) Perspective that is missing: Some model studies don't show invigoration in tropical convection. Is this case-dependent, or microphysics-dependent, or something else? A few simulations with a standard 2 moment WRF microphysics scheme would be a quick test that would help clarify. You don't need to reproduce the entire analysis for a bulk scheme – just a few key fields.

4) The polluted cases have different distributions of surface precipitation (Fig. 2) but how different are the total surface accumulations?

Other Comments:

5) Please explain the reason for the model feature in the CFAD at 5-8 km where there is significant difference from the radar.

6) Line 175: 'may represent'. The model can tell you whether this is the case.

7) Lines 278 and 296: 'exact' is too strong for a model

8) Line 293: 'overall effect' (of what?)

9) Fig. 7: As mentioned above, the third column is apparently only at ZTL but the caption says 'over all cloudy areas'. Throughout the paper please check figure labels so that this is unambiguous.

10) Line 332: 'extremely' should be removed.

11) Some shortened labels are a bit strange ('pollu', 'depo', 'frez'). 'pol', 'dep', 'frz', 'sub' would be better.

12) Please mark the approximate height of the ZTL on the profile figures. I think it is only on Pg 14 that it is stated that ZTL is at ∼ 4.8 km.

---

## Referee Comment (RC2) · Anonymous Referee #2 · 8 Feb 2017

This manuscript focuses on how changes to the warm-phase component of a convective cloud is modified by changes to aerosol amount. They simulate a convective cloud system from a field project over the Marshall Islands using WRF. The most interesting aspect of this work is the use of the what they call the VCOG or the combination of the surrounding air velocity and the effective terminal velocity of hydrometeors. This work is well written, easy to read and to follow, the results follow clearly from their analysis and the figures are well chosen and presented.

My recommendation to accept his work with minor revisions

Main comments: 1) It would also be beneficial to see a discussion about the applicability of this case to other convective systems. How generalizable are these results?

2) The discussion of Figure 2 states that they model CFAD captures the vertical structure and magnitude of the observed CFAD reasonably well. I agree that the highest probabilities (dark red) do look similar between the observations and the model. However, what is the source of the strange peak in the modeled CFAD between 5-7 km that is not present in the observations?

3) For Figure 8, what is the source of the smooth nature of the clean curve in 8a and c vs. the more variable semipolluted and polluted curves?

Line by line comments: Line 164: "Aerosol effects on clouds' macroscale" is strange wording. Perhaps "Aerosol effects on macroscale cloud properties" would be a better section header.

Line 166-167: "vertical profiles of a cloud fraction" - I don't think you need the "a"

Line 167-168: What is a "Voxel"? I have never heard this term before.

Line 184: Voxel again... once it's introduced earlier this would be fine.

Line 296: VCOG - "COG" should be subscripted

Line 298: VCOG - "COG" should be subscripted

Line 299: VCOG - "COG" should be subscripted

Line 300: VCOG - "COG" should be subscripted

Figure comments: General: Most of the figures appear to have bolded text for the axes and color bars, for some (specifically the color bar on Figure 3 b,d,f) is blurry due to this. The text in figure 7 is also very blurry.

Figure 4b) The insert is hard to see, perhaps switch the location of the legend and the zoomed in insert so that the insert can be made larger.

Figure 5a,b,c) including the ZTL as a dashed line on these figures would be helpful.

———————————————

---

## Author Comment (AC1) · 3 May 2017

*A reply to the review of* "How do changes in warm-phase microphysics affect deep convective clouds?" *by* Qian Chen et al.

**Referee #1**

We thank the reviewer for the efforts and beneficial comments that helped us improve the paper and present a clearer and more complete study. Please find below a point by point reply to all comments (replies in blue):

This manuscript presents an analysis of aerosol effects on tropical deep convection using the WRF model with detailed bin microphysics. It emphasis the zero degree level as explaining the system wide behavior. It adds analysis of the velocity center of gravity (VCOG) to help explain the aerosol invigoration of deep convection, which is a hotly debated topic in the cloud physics/dynamics community.

**Answer**: We thank the reviewer for the important comments that helped us focus the message of this paper. Indeed invigoration is - a hotly debated topic - and in this work we wanted to highlight some key processes that can explain aerosol effects on deep convective systems from (mostly) the warm perspective, which we understand better. Moreover, by focusing on the zero temperature level (ZTL) as a reference for fluxes in both directions we could show that elevated aerosol level boosts the system's dynamics in both directions (upward and downward), such that the mass fluxes up and down that cross the ZTL are much larger for the polluted system. Hence, the observed invigoration effect can be viewed as the residual of the larger fluxes that act to partly cancel each other. As the reviewer stated throughout the review, we show that the main drivers for the enhanced dynamics are aerosol enhancement of the diffusion processes and aerosol enhancement of hydrometeors' mobility. To keep the paper focused and accessible we provide a domain-perspective study in which we analyze mostly mean values of key processes.

I have a few comments that I think will make the paper much more persuasive.

1) A counter argument to invigoration is that the added mass loading reduces the buoyancy and offsets the higher condensation/latent heating. Please show analysis (e.g., profile time series) of the components of the buoyancy (thetav' and mass

loading) for all cases. Mass loading should be broken down for the different hydrometeors. This is important in explaining why the added mass loading does not seem to make much difference. Can it be tied to the hydrometeor mobility? This perspective would make the paper much stronger.

**Answer**: We thank the reviewer for this comment. Indeed as the reviewer suggested water loading plays an important role in counter balancing the total buoyancy, and droplet mobility explains part of the enhanced fluxes and overall trend.

Following this comment and for explaining it better we added a new paragraph (including a new figure) to section 3.3: "*The enhanced gain of water and ice mass in the polluted runs yielded higher mass loading that acted to reduce the clouds' buoyancy. The upper panel of Figure 7 shows the vertical profiles of the mean buoyancy (total B and components: BT – thermal, BV – vapor and WL – water loading) for the domains' cloudy voxels (between 4 and 12 h of the simulation). Indeed, the water loading played an important role and as expected there was a 'payment', once the polluted clouds got thicker with more gained liquid and ice mass that was transported higher in the atmosphere, the added water loading acted to counter-balance the overall buoyancy. Figure 7a shows that the total B profiles switch between smaller values (more negative) for the polluted runs compared to the clean run in low altitudes (from 2.4 to 6.7 km) to larger values at higher levels (above 6.7 km). In all levels, BT was larger in the polluted case (or equal near the freezing level) and the water loading was the smallest (less negative WL buoyancy component) in the clean run, from 1.5 to 8.5 km. The lower panel of Figure 7 shows the vertical profiles of the mean water loading components for the different types of cloud particles. Note it is different from the profiles of total mass shown in Figure 6 as it presents the mean B values and hence it is influenced by the cloud coverage in each level. It shows significant increase in the water loading (smaller WT buoyancy component) for the polluted runs in all but the snow hydrometeors. Moreover, it reveals that larger rain content, that was likely to originate higher in the clouds can explain about half of the extra water loading in the polluted cases in the lower part of the clouds.*"

[Figure]

Fig. 7: Vertical profiles of (a) mean buoyancy averaged over cloudy regions between 4-12 hours of simulation and its components: (b) thermal term (BT), (c) water vapor term (BV), and (d) water loading term (WL); (e-i) vertical profiles of WL for cloud droplet (e), raindrop (f), ice crystal (g), snow (h), graupel (i). The black dashed lines denote the ZTL (~4.8 km).

2) There is emphasis in the tone of the paper (even the title) on understanding the warm phase and transport of water across the ZTL and how that affects the mixed phase above. But I didn't find this persuasive. E.g., On page 14 it is stated: "most of the cloud drops that grew by condensation above the ZTL originated in the warm part of the clouds. Therefore, we can clearly state that an important part of the enlarged liquid mass generated below the ZTL in the more polluted cases (as shown in Figure 6a,b) was transported upward with the stronger updrafts". However, a more straightforward interpretation is that the stronger updrafts, which extend above the ZTL, generate more cooling, higher supersaturation and therefore more condensate. The 'transport' argument doesn't seem particularly clear or strong. On this point the figures need some work (see # 9 below). My suggestion is to either strengthen the argument of why a stronger mass flux at ZTL necessarily carries up through the depth of the cloud, or alternatively remove the emphasis.

**Answer**: We thank the reviewer for this important comment that again helped us telling the paper's story better. We have added a paragraph, discussing mass flux vs. local production above the ZTL into section 3.4: "*While the temporal evolution of the overall fluxes crossing the ZTL is not dramatically different for the three runs (mean values of ~4.17, 4.77 and 3.46×10$^6$ kg s$^{-1}$ for the polluted, semi-polluted, and clean run, respectively), when the fluxes are observed separately according to their sign, a different view emerges. All flux trends (up and down) were dramatically enhanced in the polluted cases. The upward liquid mass fluxes in the semi-polluted and polluted cases were 4–9 and 8–21 times larger than those in the clean simulation (with mean values of 3.9, 1.8 and 0.32×10$^6$ kg s$^{-1}$ for the polluted, semi-polluted, and clean run). These trends are controlled by the enlarged upward motion area (as shown in Figure 9d), the larger COG velocity (Figure 8d), and the increased liquid water loading. These findings clearly demonstrate the enhanced cloud mass transport to upper levels under more polluted conditions. What is the average ratio between the mass that was transported up from below the ZTL and the mass that was produced locally above the ZTL, in the different runs? This mass ratio is defined as*
$\mu = \frac{transported\ mass\ from\ below\ the\ ZTL}{mass\ locally\ produced\ above\ the\ ZTL}$ *and it is evaluated using the mass fluxes (as shown in Figure 9). If we consider mean values over 8 h of simulation, for which changes in the mass above the ZTL are averaged out, we can use the domain's average mass fluxes across the ZTL to estimate the mass ratio as*
$\mu \approx \frac{mass\ flux\ upward}{net\ mass\ flux}$. *The mass-fraction $\mu$ is much higher in the polluted as compared to the clean run (around 0.94, 0.38 and 0.09 in the polluted, semi-polluted, and clean cases).*"

In addition we have clarified the text describing processes controlling the mass profiles in section 3.3: "*It is important to note that among all of the processes, condensation contributed most to the mass gain. It peaked at 2.4 km for all three runs. Moreover, even above the ZTL, condensation still dominated the cloud mass gain compared to depositional growth, for all three runs (Figure 5a vs. 5b). Since the droplet-nucleation process was negligible above the ZTL in the present study (accounting for ~5% of the total nucleated drop mass), most of the cloud drops that*

*grew by condensation above the ZTL originated in the warm part of the clouds. The mass gained by the enhanced condensation in the polluted runs was further boosted by the stronger updrafts that increased the supersaturation. Moreover, having better droplet mobility (see Figure 6g) further implies that a significant part of the enlarged liquid mass generated below the ZTL in the more polluted cases (as shown in Figure 6a,b) was transported upward by the stronger updrafts (Figure 3h) and continued growing via condensation at altitudes above the ZTL."*

3) Perspective that is missing: Some model studies don't show invigoration in tropical convection. Is this case-dependent, or microphysics-dependent, or something else? A few simulations with a standard 2 moment WRF microphysics scheme would be a quick test that would help clarify. You don't need to reproduce the entire analysis for a bulk scheme – just a few key fields.

**Answer**: In this study we hoped to avoid giving a critical view on the bulk vs. bin schemes and why bulk schemes that are commonly using saturation-adjustment are limited in their ability to capture invigoration processes. Recent studies of bulk vs. bin schemes comparisons are accumulating, and many of them show how in-essence saturation-adjustment (Tao et al., 1989) mimics polluted runs even for low aerosol concentration. It is caused by neglecting the time it takes to consume the supersaturation and therefore the bulk schemes dictate excellent condensation efficiency for all runs with limited sensitivity to aerosol concentration (Lebo and Seinfeld 2011; Lebo et al. 2012; Khain et al., 2015; Heiblum et al, 2016). Other comparison studies indicated of bulk schemes limitation of the prescribed hydrometeors size distribution and autoconversion parameterization (Ovchinnikov et al., 2014). Khain et al., (2009, 2015) showed that schemes that prescribe the drop concentration cannot capture correctly the sensitivity of cloud and rain processes to changes in aerosols amount.

We agree that the generality of every numerical simulation study should be questioned. However, many recent numerical studies do show similar invigoration trends for deep tropical convective clouds, from a single cloud to a squall line system (Storer et al. 2013; Cui et al., 2011; Fan et al., 2013; Khain et al., 2008; Li et al., 2013; Tao et al., 2007; Tao and Li, 2016). Following this comment, we have added a part discussing

these issues into section 3.2: *"Our results agree with previous numerical studies that reported an aerosol invigoration effect of tropical deep convective clouds (Cui et al., 2011; Fan et al., 2013; Khain et al., 2008; Li et al., 2013; Tao and Li, 2016; Tao et al., 2007). However other numerical studies showed no clear evidence for this effect or even an opposite effect (Lee and Feingold, 2010; Morrison and Grabowski, 2011, 2013). The reasons behind those differences were examined in previous studies that showed the lower sensitivity of cloud and rain processes in bulk schemes to aerosol concentration (Khain et al., 2009, 2015; Lebo and Seinfeld, 2011; Lebo et al., 2012; Heiblum et al., 2016)."*

4) The polluted cases have different distributions of surface precipitation (Fig. 2) but how different are the total surface accumulations?

**Answer**: Thank you for this comment. The time evolution of domain-mean accumulated surface precipitation has been added to Figure 9g (Figure 8g in the previous version, see below), which shows the larger amount of accumulated rain in more polluted cases. We revised the corresponding description at the end of section 3.4: *"Similar to the upward transport, the downward transport of cloud mass from subzero temperature levels to the warm environment (Figure 9e), the mean surface rain rate and accumulated precipitation (Figure 9g) were also larger in the more polluted cases."*

[Figure]

**Fig. 9: Time series of (a) net cloud mass fluxes, and (b) area fraction over entire cloudy area crossing the ZTL for clean (blue curve), semi-polluted (green curve), and polluted (red curve) experiments. (c, d) and (e, f) are similar to (a, b), but for upward ($V_{COG} > 0$) and downward ($V_{COG} < 0$) motion areas, respectively. (g) is time evolution of domain-averaged rain rate (solid lines) and accumulated surface rain (dashed lines). (h) is time evolution of rainy area fraction at the surface.**

Other Comments:

5) Please explain the reason for the model feature in the CFAD at 5-8 km where there is significant difference from the radar.

**Answer**: The model overestimates the reflectivity values above 4.8 km (the environmental ZTL) and there is a sharp decrease in reflectivity below it. This can be attributed to overestimation of big ice particles above 5 km (mostly graupels, but snow particles as well). It is a result of the relatively simple melting scheme used by the model that allows immediate melting of ice particles when falling across the ZTL.

Hence, large graupel and ice particles (as indicated by their large effective terminal velocity presented in Figures 6j,k) melt while crossing the ZTL and breakup immediately into smaller drops. Part of these drops is pushed upward by the updraft and contributes to the additional growth by riming of the graupels. And indeed 70% of the mass located above 5 km in voxels with reflectivity values higher than 35 dBZ are graupel particles. So there is an overestimation of big graupel (and snow) particles above 5 km. Below the ZTL there is a sharp decrease in reflectivity because the drops are smaller (compared to the large graupel particles above the ZTL) and they fall faster so their concentration in the volume is reduced (hence form a reduced reflectivity). Thank to this remark we added the text (sections 3.1 and 3.3) parts that highlights the limitation of the melting scheme and explain the feedbacks caused by it.

The additions to section 3.1: *"There is an overestimation of the modeled reflectivity above the ZTL (4.8 km) compared to the observed one. It can be explained by an overestimation of large ice hydrometeors (mostly graupel, but snow particles as well) above the ZTL. This is due to feedbacks caused by the simple melting scheme used by the model (see section 3.3)."*

The changes in section 3.3: *"Note that the simple melting scheme used by the model allowed immediate melting of ice particles while crossing the ZTL. The resulted drops formed by the melting of big graupel (and snow) particles broke up immediately into smaller drops and part of them was carried up again by the updraft and froze by riming. So there may be an overestimation of big graupel particles above the ZTL (as shown in Figure 2b)."*

6) Line 175: 'may represent'. The model can tell you whether this is the case.

**Answer**: Thank you. We changed the statement to 'includes': "We note that the cloud fraction below 1 km *includes* precipitating regions."

7) Lines 278 and 296: 'exact' is too strong for a model

**Answer**: Thank you for this comment. The word 'exact' was used in the context of the definition of the effective terminal velocity. Nevertheless, in order to avoid confusion it was deleted from the revised text.

8) Line 293: 'overall effect' (of what?)

**Answer**: It is the 'overall aerosol effect'. We added it in the sentence: "What roles do the two characteristic velocities play in the overall *aerosol* effect?"

9) Fig. 7: As mentioned above, the third column is apparently only at ZTL but the caption says 'over all cloudy areas'. Throughout the paper please check figure labels so that this is unambiguous.

**Answer**: Thank you for this comment. The labels of Figures 7a-c (now Figures 8a-c) have been corrected to "$V_{COG}$ *over cloudy area at ZTL*", "*w over cloudy area at ZTL*", and "*|η| over cloudy area at ZTL*", respectively.

10) Line 332: 'extremely' should be removed.

**Answer**: Thank you. Removed.

11) Some shortened labels are a bit strange ('pollu', 'depo', 'frez'). 'pol', 'dep', 'frz', 'sub' would be better.

**Answer**: Thank you. We have changed the abbreviation in all the figures as suggested by the reviewer.

12) Please mark the approximate height of the ZTL on the profile figures. I think it is only on Pg 14 that it is stated that ZTL is at ~ 4.8 km.

**Answer**: Thank you. Lines that mark the ZTL were added to Figs. 5, 6, and 7.

**References**

Cui, Z., Davies, S., Carslaw, K. S., and Blyth, A. M.: The response of precipitation to aerosol through riming and melting in deep convective clouds, Atmos. Chem. Phys., 11, 3495-3510, 10.5194/acp-11-3495-2011, 2011.

Fan, J., Leung, L. R., Rosenfeld, D., Chen, Q., Li, Z., Zhang, J., and Yan, H.: Microphysical effects determine macrophysical response for aerosol impacts on deep convective clouds, Proc. Natl. Acad. Sci., 110, E4581-E4590, 10.1073/pnas.1316830110, 2013.

Heiblum, R. H., Altaratz, O., Koren, I., Feingold, G., Kostinski, A. B., Khain, A. P., Ovchinnikov, M., Fredj, E., Dagan, G., Pinto, L., Yaish, R., and Chen, Q.: Characterization of cumulus cloud fields using trajectories in the center-of-gravity vs. water mass phase space. Part II: Aerosol effects on warm convective clouds, J. Geophys. Res., 6356-6373, 10.1002/2015JD024193, 2016.

Khain, A. P., Beheng, K. D., Heymsfield, A., Korolev, A., Krichak, S. O., Levin, Z., Pinsky, M., Phillips, V., Prabhakaran, T., Teller, A., van den Heever, S. C., and Yano, J. I.: Representation of microphysical processes in cloud-resolving models: spectral (bin) microphysics vs. bulk parameterization, Rev. Geophys., 2014RG000468, 10.1002/2014RG000468, 2015.

Khain, A. P., BenMoshe, N., and Pokrovsky, A.: Factors determining the impact of aerosols on surface precipitation from clouds: An attempt at classification, J. Atmos. Sci., 65, 1721-1748, 2008.

Khain, A. P., Leung, L. R., Lynn, B., and Ghan, S.: Effects of aerosols on the dynamics and microphysics of squall lines simulated by spectral bin and bulk parameterization schemes, J. Geophys. Res., 114, D22203, 10.1029/2009jd011902, 2009.

Lebo, Z. J., Morrison, H., and Seinfeld, J. H.: Are simulated aerosol-induced effects on deep convective clouds strongly dependent on saturation adjustment?, Atmos. Chem. Phys. Discuss., 12, 10059-10114, 10.5194/acpd-12-10059-2012, 2012.

Lebo, Z. J., and Seinfeld, J. H.: Theoretical basis for convective invigoration due to increased aerosol concentration, Atmos. Chem. Phys., 11, 5407-5429, 10.5194/acp-11-5407-2011, 2011.

Lee, S.-S., and Feingold, G.: Precipitating cloud-system response to aerosol perturbations, Geophys. Res. Lett., 37, L23806, 10.1029/2010gl045596, 2010.

Li, X., Tao, W., Masunaga, H., Gao, G., and Zeng, X.: Aerosol Effects on Cumulus Congestus Population over the Tropical Pacific: A Cloud-Resolving Modeling Study, J. Meteor. Soc. Japan., 91, 817-833, 2013.

Morrison, H., and Grabowski, W. W.: Cloud-system resolving model simulations of aerosol indirect effects on tropical deep convection and its thermodynamic environment, Atmos. Chem. Phys., 11, 10503-10523, 10.5194/acp-11-10503-2011, 2011.

Morrison, H., and Grabowski, W. W.: Response of Tropical Deep Convection to Localized Heating Perturbations: Implications for Aerosol-Induced Convective Invigoration, J. Atmos. Sci., 70, 3533-3555, 10.1175/jas-d-13-027.1, 2013.

Ovchinnikov, M., Ackerman, A. S., Avramov, A., Cheng, A., Fan, J., Fridlind, A. M., Ghan, S., Harrington, J., Hoose, C., Korolev, A., McFarquhar, G. M., Morrison, H., Paukert, M., Savre, J., Shipway, B. J., Shupe, M. D., Solomon, A., and Sulia, K.: Intercomparison of large-eddy simulations of Arctic mixed-phase clouds: Importance of ice size distribution assumptions, J. Adv. Model. Earth Syst., 6, 223-248, 10.1002/2013MS000282, 2014.

Storer, R. L., and van den Heever, S. C.: Microphysical processes evident in aerosol forcing of tropical deep convective clouds, J. Atmos. Sci., 70, 430-446, 10.1175/jas-d-12-076.1, 2013.

Tao, W.-K., Simpson, J., and McCumber, M.: An Ice-Water Saturation Adjustment, Mon. Wea. Rev., 117, 231-235, 1989.

Tao, W.-K., and Li, X.: The relationship between latent heating, vertical velocity, and precipitation processes: The impact of aerosols on precipitation in organized deep convective systems, J.

Geophys. Res., 121, 6299–6320, 10.1002/2015JD024267, 2016.

Tao, W.-K., Li, X., Khain, A., Matsui, T., Lang, S., and Simpson, J.: Role of atmospheric aerosol concentration on deep convective precipitation: Cloud-resolving model simulations, J. Geophys. Res., 112, D24S18, 2007.

---

## Author Comment (AC2) · 3 May 2017

We thank the reviewer for his thoughtful comments that helped us improve the paper. Please find below a point by point reply to all comments (replies in blue):

This manuscript focuses on how changes to the warm-phase component of a convective cloud is modified by changes to aerosol amount. They simulate a convective cloud system from a field project over the Marshall Islands using WRF. The most interesting aspect of this work is the use of the what they call the VCOG or the combination of the surrounding air velocity and the effective terminal velocity of hydrometeors. This work is well written, easy to read and to follow, the results follow clearly from their analysis and the figures are well chosen and presented.

My recommendation to accept his work with minor revisions

**Answer**: We are glad that the reviewer found our manuscript interesting and well presented.

Main comments: 1) It would also be beneficial to see a discussion about the applicability of this case to other convective systems. How generalizable are these results?

**Answer**: Thank you for this comment. Indeed it is very important to discuss the generality of our results. More and more studies (both observational and numerical ones) are accumulating showing clear evidences for invigoration of deep convective clouds. For example numerical studies (using both bin and bulk schemes) of single tropical cloud up to mesoscale convective system like squall line (Sarangi et al, 2015; Storer et al. 2013; Cui et al., 2011; Fan et al., 2013; Khain et al., 2008; Li et al., 2013; Tao et al., 2007; Tao and Li, 2016) and observational studies (Sarangi et al, 2017; Jiang et al, 2016; Storer et al., 2014; Yan et al, 2014; Heiblum et al., 2012; Koren et al., 2005, 2010; Andreae et al., 2004).

We are aware however, that there were numerical studies that showed no clear

evidence or even an opposite aerosol effect. They all used bulk microphysical schemes (Lee and Feingold, 2010; Morrison and Grabowski, 2011; Morrison and Grabowski, 2013). We think that due to some inherent properties of the common bulk schemes, such model experiment are significantly less sensitive to aerosol effect. To name some of the main limitations: recent studies of bulk vs. bin schemes comparison show how in-essence saturation-adjustment (Tao et al., 1989) mimics polluted runs even for low aerosol concentration. It is caused by neglecting the time it takes to consume the supersaturation and therefore the bulk schemes dictate excellent condensation efficiency for all runs with limited sensitivity to aerosol concentration (Lebo and Seinfeld 2011; Lebo et al. 2012; Khain et al., 2015; Heiblum et al, 2016). Other comparison studies indicated of bulk schemes limitation of the prescribed hydrometeors size distribution and autoconversion parameterization (Ovchinnikov et al., 2014). Khain et al., (2009, 2015) showed that schemes that prescribe the drop concentration cannot capture correctly the sensitivity of cloud and rain processes to changes in aerosols amount.

To present these points and to make it clearer we have revised Section 3.2 as follows:

*"Our results agree with previous numerical studies that reported an aerosol invigoration effect of tropical deep convective clouds (Cui et al., 2011; Fan et al., 2013; Khain et al., 2008; Li et al., 2013; Tao and Li, 2016; Tao et al., 2007). However other numerical studies showed no clear evidence for this effect or even an opposite effect (Lee and Feingold, 2010; Morrison and Grabowski, 2011, 2013). The reasons behind those differences were examined in previous studies that showed the lower sensitivity of cloud and rain processes in bulk schemes to aerosol concentration (Khain et al., 2009, 2015; Lebo and Seinfeld, 2011; Lebo et al., 2012; Heiblum et al., 2016)."*

2) The discussion of Figure 2 states that they model CFAD captures the vertical structure and magnitude of the observed CFAD reasonably well. I agree that the highest probabilities (dark red) do look similar between the observations and the model. However, what is the source of the strange peak in the modeled CFAD between 5-7 km that is not present in the observations?

**Answer**: The model overestimates the reflectivity values above 4.8 km (the

environmental ZTL) and there is a sharp decrease in reflectivity below it. This can be attributed to overestimation of big ice particles above 5 km (mostly graupels, but snow particles as well). It is a result of the relatively simple melting scheme used by the model that allows immediate melting of ice particles when falling across the ZTL. So large graupel and ice particles (as indicated by their large effective terminal velocity presented in Figures 6j,k) melt while crossing the ZTL and breakup immediately into smaller drops. Part of these drops is pushed upward by the updraft and contributes to the additional growth by riming of the graupels. And indeed 70% of the mass located above 5 km in voxels with reflectivity values higher than 35 dBZ are graupel particles. So there is an overestimation of big graupel (and snow) particles above 5 km. Below the ZTL there is a sharp decrease in reflectivity because the drops are smaller (compared to the large graupel particles above the ZTL) and they fall faster so their concentration in the volume is reduced (hence form a reduced reflectivity). Thank to this remark we added the text (sections 3.1 and 3.3) parts that highlights the limitation of the melting scheme and explain the feedbacks caused by it.

The additions to section 3.1: *"There is an overestimation of the modeled reflectivity above the ZTL (4.8 km) compared to the observed one. It can be explained by an overestimation of large ice hydrometeors (mostly graupel, but snow particles as well) above the ZTL. This is due to feedbacks caused by the simple melting scheme used by the model (see section 3.3)."*

The changes in section 3.3: *"Note that the simple melting scheme used by the model allowed immediate melting of ice particles while crossing the ZTL. The resulted drops formed by the melting of big graupel (and snow) particles broke up immediately into smaller drops and part of them was carried up again by the updraft and froze by riming. So there may be an overestimation of big grauple particles above the ZTL (as shown in Figure 2b)."*

3) For Figure 8, what is the source of the smooth nature of the clean curve in 8a and c vs. the more variable semipolluted and polluted curves?

**Answer**: Thank you for this great observation. Following this comment, we revised section 3.4 to point out and explain this variance:

*"This impacted the variance of the mass-flux, which was larger in the more polluted*

*cases (Figure 9c). The increased variance is driven by the enhancement of the fields'*
*dynamics by aerosol, as shown throughout this study. Polluted clouds exhibited larger*
*updrafts with larger variance (as shown in Figure 8), larger updraft area (Figure 9d)*
*and larger mass fluxes, all of which tend to increase the variance in the upward*
*mass-flux*".

Line by line comments: Line 164: "Aerosol effects on clouds' macroscale" is strange wording. Perhaps "Aerosol effects on macroscale cloud properties" would be a better section header.

**Answer**: Thank you. We changed it to "*Aerosol effects on clouds' macrophysical properties*".

Line 166-167: "vertical profiles of a cloud fraction" - I don't think you need the "a"

**Answer**: Thank you. Corrected.

Line 167-168: What is a "Voxel"? I have never heard this term before.

Line 184: Voxel again... once it's introduced earlier this would be fine.

**Answer**: It is the abbreviation of volume pixel, which is the smallest unit of three-dimensional grid-space. Here it means a grid volume. Explanation has been added in the text: "Figure 3a,c,e shows the evolution of the vertical profiles of a cloud fraction for the three runs (calculated as the ratio between the number of cloudy *volume pixels (voxels)*, i.e., total condensate exceeding 0.01 g kg$^{-1}$, at each vertical level and the total horizontal number of voxels)."

Line 296: VCOG - "COG" should be subscripted

Line 298: VCOG - "COG" should be subscripted

Line 299: VCOG - "COG" should be subscripted

Line 300: VCOG - "COG" should be subscripted

**Answer**: Thank you. Corrected.

Figure comments: General: Most of the figures appear to have bolded text for the axes and color bars, for some (specifically the color bar on Figure 3 b,d,f) is blurry due to this. The text in figure 7 is also very blurry.

**Answer**: Thank you. Revised.

Figure 4b) The insert is hard to see, perhaps switch the location of the legend and the zoomed in insert so that the insert can be made larger.

**Answer**: Thank you. Revised.

Figure 5a,b,c) including the ZTL as a dashed line on these figures would be helpful.
**Answer**: Thank you. The ZTL lines have been added into figures 5,6 and 7.

**References**

Andreae, M. O., Rosenfeld, D., Artaxo, P., Costa, A. A., Frank, G. P., Longo, K. M., and Silva-Dias, M. A. F.: Smoking rain clouds over the Amazon, Science, 303, 1337, 2004.

Cui, Z., Davies, S., Carslaw, K. S., and Blyth, A. M.: The response of precipitation to aerosol through riming and melting in deep convective clouds, Atmos. Chem. Phys., 11, 3495-3510, 10.5194/acp-11-3495-2011, 2011.

Fan, J., Leung, L. R., Rosenfeld, D., Chen, Q., Li, Z., Zhang, J., and Yan, H.: Microphysical effects determine macrophysical response for aerosol impacts on deep convective clouds, Proc. Natl. Acad. Sci., 110, E4581-E4590, 10.1073/pnas.1316830110, 2013.

Heiblum, R. H., Koren, I., and Altaratz, O.: New evidence of cloud invigoration from TRMM measurements of rain center of gravity, Geophys. Res. Lett., 39, L08803, 10.1029/2012gl051158, 2012.

Heiblum, R. H., Altaratz, O., Koren, I., Feingold, G., Kostinski, A. B., Khain, A. P., Ovchinnikov, M., Fredj, E., Dagan, G., Pinto, L., Yaish, R., and Chen, Q.: Characterization of cumulus cloud fields using trajectories in the center-of-gravity vs. water mass phase space. Part II: Aerosol effects on warm convective clouds, J. Geophys. Res., 6356-6373, 10.1002/2015JD024193, 2016.

Jiang, M., Li, Z., Wan, B., and Cribb, M.: Impact of aerosols on precipitation from deep convective clouds in eastern China, J. Geophys. Res. Atmos., 121, 9607-9620, 10.1002/2015JD024246,

2016.

Khain, A. P., Beheng, K. D., Heymsfield, A., Korolev, A., Krichak, S. O., Levin, Z., Pinsky, M., Phillips, V., Prabhakaran, T., Teller, A., van den Heever, S. C., and Yano, J. I.: Representation of microphysical processes in cloud-resolving models: spectral (bin) microphysics vs. bulk parameterization, Rev. Geophys., 2014RG000468, 10.1002/2014RG000468, 2015.

Khain, A. P., BenMoshe, N., and Pokrovsky, A.: Factors determining the impact of aerosols on surface precipitation from clouds: An attempt at classification, J. Atmos. Sci., 65, 1721-1748, 2008.

Khain, A. P., Leung, L. R., Lynn, B., and Ghan, S.: Effects of aerosols on the dynamics and microphysics of squall lines simulated by spectral bin and bulk parameterization schemes, J. Geophys. Res., 114, D22203, 10.1029/2009jd011902, 2009.

Koren, I., Feingold, G., and Remer, L. A.: The invigoration of deep convective clouds over the Atlantic: aerosol effect, meteorology or retrieval artifact?, Atmos. Chem. Phys., 10, 8855-8872, 10.5194/acp-10-8855-2010, 2010.

Koren, I., Kaufman, Y. J., Rosenfeld, D., Remer, L. A., and Rudich, Y.: Aerosol invigoration and restructuring of Atlantic convective clouds, Geophys. Res. Lett, 32, L14828, 10.1029/2005gl023187, 2005.

Lebo, Z. J., Morrison, H., and Seinfeld, J. H.: Are simulated aerosol-induced effects on deep convective clouds strongly dependent on saturation adjustment?, Atmos. Chem. Phys. Discuss., 12, 10059-10114, 10.5194/acpd-12-10059-2012, 2012.

Lebo, Z. J., and Seinfeld, J. H.: Theoretical basis for convective invigoration due to increased aerosol concentration, Atmos. Chem. Phys., 11, 5407-5429, 10.5194/acp-11-5407-2011, 2011.

Lee, S.-S., and Feingold, G.: Precipitating cloud-system response to aerosol perturbations, Geophys. Res. Lett., 37, L23806, 10.1029/2010gl045596, 2010.

Li, X., Tao, W., Masunaga, H., Gao, G., and Zeng, X.: Aerosol Effects on Cumulus Congestus Population over the Tropical Pacific: A Cloud-Resolving Modeling Study, J. Meteor. Soc. Japan., 91, 817-833, 2013.

Morrison, H., and Grabowski, W. W.: Cloud-system resolving model simulations of aerosol indirect effects on tropical deep convection and its thermodynamic environment, Atmos. Chem. Phys., 11, 10503-10523, 10.5194/acp-11-10503-2011, 2011.

Morrison, H., and Grabowski, W. W.: Response of Tropical Deep Convection to Localized Heating Perturbations: Implications for Aerosol-Induced Convective Invigoration, J. Atmos. Sci., 70, 3533-3555, 10.1175/jas-d-13-027.1, 2013.

Ovchinnikov, M., Ackerman, A. S., Avramov, A., Cheng, A., Fan, J., Fridlind, A. M., Ghan, S., Harrington, J., Hoose, C., Korolev, A., McFarquhar, G. M., Morrison, H., Paukert, M., Savre, J., Shipway, B. J., Shupe, M. D., Solomon, A., and Sulia, K.: Intercomparison of large-eddy simulations of Arctic mixed-phase clouds: Importance of ice size distribution assumptions, J. Adv. Model. Earth Syst., 6, 223-248, 10.1002/2013MS000282, 2014.

Sarangi, C., Tripathi, S. N., Kanawade, V. P., Koren, I., and Pai, D. S.: Investigation of the aerosol–cloud–rainfall association over the Indian summer monsoon region, Atmos. Chem. Phys., 17, 5185-5204, doi:10.5194/acp-17-5185-2017, 2017.

Sarangi, C., Tripathi, S. N., Tripathi, S., and Barth, M. C.: Aerosolcloud associations over Gangetic Basin during a typical monsoon depression event using WRF-Chem simulation, J. Geophys. Res.-Atmos., 120, 10974–10995, 2015

Storer, R. L., and van den Heever, S. C.: Microphysical processes evident in aerosol forcing of tropical

deep convective clouds, J. Atmos. Sci., 70, 430-446, 10.1175/jas-d-12-076.1, 2013.

Storer, R. L., van den Heever, S. C., and L'Ecuyer, T. S.: Observations of aerosol induced convective invigoration in the tropical East Atlantic, J. Geophys. Res., 2013JD020272, 10.1002/2013jd020272, 2014.

Tao, W.-K., Simpson, J., and McCumber, M.: An Ice-Water Saturation Adjustment, Mon. Wea. Rev., 117, 231-235, 1989.

Tao, W.-K., and Li, X.: The relationship between latent heating, vertical velocity, and precipitation processes: The impact of aerosols on precipitation in organized deep convective systems, J. Geophys. Res., 121, 6299–6320, 10.1002/2015JD024267, 2016.

Tao, W.-K., Li, X., Khain, A., Matsui, T., Lang, S., and Simpson, J.: Role of atmospheric aerosol concentration on deep convective precipitation: Cloud-resolving model simulations, J. Geophys. Res., 112, D24S18, 2007.

Yan, H., Li, Z., Huang, J., Cribb, M., and Liu, J.: Long-term aerosol-mediated changes in cloud radiative forcing of deep clouds at the top and bottom of the atmosphere over the Southern Great Plains, Atmos. Chem. Phys., 14, 7113-7124, doi:10.5194/acp-14-7113-2014, 2014.

---

## Author Response (AR2)

July 6, 2017

Dear Prof. Facchini

Thank you so much for selecting our paper and acting as the ACP editor for it.

5    As instructed we have revised the paper, following the reviewer's suggestions to incorporate the new findings into the abstract and summary.

We have modified the abstract and added the following section: "Increased condensation efficiency of cloud droplets governed the gain in mass below the ZTL, while both enhanced condensational and depositional growth led to increased mass above it. The enhanced mass loading above the ZTL acted to

10   reduce the clouds' buoyancy while the thermal buoyancy (driven by the enhanced latent heat release) increased in the polluted runs. The overall effect showed an increased upward transport (across the ZTL) of liquid water, driven by both larger updrafts and larger droplet mobility."

In the summary part of the paper we further added: "We show that there is a "payment" for the enhanced mass production and transport in the polluted runs. The larger mass loading act to reduce the

15   buoyancy while the enhanced latent heat release act to increase it. The enhanced mass flux upward and the enhanced mass production above the ZTL yield enhancement of the mass flux down (partly manifested as enhanced rain that explains much of the increase in mass loading below the ZTL)."

These additions indeed provide a fuller perspective of the papers findings.

20   Thank you and best wishes,

Ilan Koren

[revised manuscript text omitted]